# CompoundPiece: Evaluating and Improving Decompounding Performance of Language Models

**Benjamin Minixhofer** [1]   **Jonas Pfeiffer** [†2]   **Ivan Vulić** [†1]

[1]University of Cambridge   [2]Google DeepMind

## Abstract

While many languages possess processes of joining two or more words to create *compound* words, previous studies have been typically limited only to languages with excessively productive compound formation (e.g., German, Dutch) and there is no public dataset containing compound *and* non-compound words across a large number of languages. In this work, we systematically study *decompounding*, the task of splitting compound words into their constituents, at a wide scale. We first address the data gap by introducing a dataset of 255k compound and non-compound words across 56 diverse languages obtained from Wiktionary. We then use this dataset to evaluate an array of Large Language Models (LLMs) on the decompounding task. We find that LLMs perform poorly, especially on words which are tokenized unfavorably by subword tokenization. We thus introduce a novel methodology to train dedicated models for decompounding. The proposed two-stage procedure relies on a fully self-supervised objective in the first stage, while the second, supervised learning stage optionally fine-tunes the model on the annotated Wiktionary data. Our self-supervised models outperform the prior best unsupervised decompounding models by 13.9% accuracy on average. Our fine-tuned models outperform all prior (language-specific) decompounding tools. Furthermore, we use our models to leverage decompounding during the creation of a subword tokenizer, which we refer to as *CompoundPiece*. CompoundPiece tokenizes compound words more favorably on average, leading to improved performance on decompounding over an otherwise equivalent model using SentencePiece tokenization.

## 1 Introduction

*Decompounding* is the task of separating compound words into their single word constituents. Decompounding is used in user-facing tools such as dictionaries and morphological analyzers (Altinok,

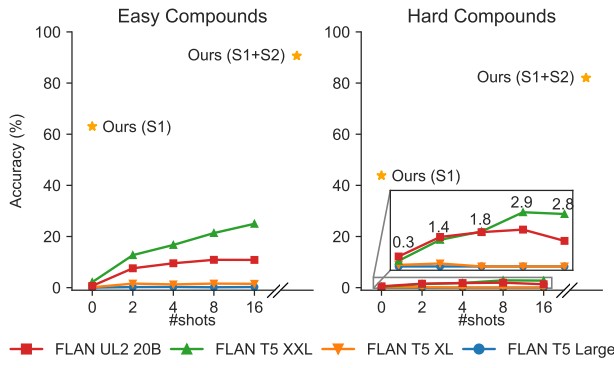

Figure 1: In-context learning performance of LLMs on compound segmentation vs. our method (§5).

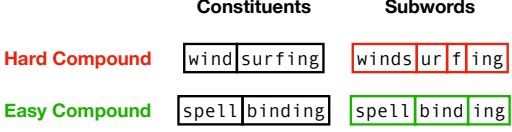

Figure 2: Examples of easy and hard compounds w.r.t. the T5 tokenizer (also used by FLAN UL2 20B).

2018). Historically, it has also been widely used as a preprocessing step for other NLP tasks, e.g. for information retrieval (Monz and De Rijke, 2002; Braschler and Ripplinger, 2004), automatic speech recognition (Adda-Decker and Adda, 2000) and machine translation (Koehn and Knight, 2003).

Decompounding can come in two similar yet different task formats: (i) *compound segmentation* and (ii) *compound normalization* (Ziering and van der Plas, 2016). Compound segmentation is the task of segmenting a word into its compound constituents, while preserving its surface form (e.g. *bridesmaid → brides + maid*). Compound normalization is the task of recovering the base form of each compound constituent (e.g. *bridesmaid → bride + maid*).[1]

Most prior work on decompounding has focused on the few languages with excessively productive

---

[†]Equal senior authorship.

[1]In morphological segmentation, segmentation and normalization are also referred to as surface-level segmentation and canonical segmentation, respectively (Cotterell et al., 2016).

compound formation such as Finnish, German and Swedish (Koehn and Knight, 2003; Shapiro, 2016; Riedl and Biemann, 2016). However, compound words occur in a large, diverse number of languages (Vogel and Scalise, 2010). Yet, datasets which annotate compounds with their segmented or normalized form sparsely exist, even in languages with high compound usage. As the first contribution of this work, we aim to address this issue by introducing a dataset of 255k compound words and their normalized form as well as non-compound words covering 56 languages obtained from Wiktionary (www.wiktionary.org).

Using our dataset, we then find that large language models (LLMs), which typically rely on subword-based tokenization (Sennrich et al., 2016; Kudo and Richardson, 2018), struggle with decompounding, as illustrated in Figure 1. Performance is especially low for compounds where subword boundaries do not coincide with compound constituent boundaries; we term compounds with this property *'hard'* compounds (Figure 2).

In order to create a more effective decompounding model, we then formulate compound segmentation and normalization as a sequence-to-sequence learning task (Sutskever et al., 2014) and train a byte-level ByT5 model (Xue et al., 2022) using a two-stage framework. In the first stage, we use a novel self-supervised hyphen-prediction objective to learn compound segmentation without any labeled data. In the second stage, we turn the model into a compound normalization model via supervised training on our Wiktionary data. In addition, we introduce a procedure to predict the segmentation of any compound word based on its normalized form, effectively making compound segmentation a subtask of normalization. Finally, we demonstrate that decompounding has real-world applications by investigating compound segmentation for language model tokenization. We apply compound segmentation as pretokenization during training of a SentencePiece tokenizer (Kudo and Richardson, 2018), which results in fewer hard compounds while incurring no extra cost during training and inference of the language model (i.e. the only extra cost occurs during creation of the tokenizer).

Our Stage 1 models outperform the best prior unsupervised models by 13.9% accuracy on average, while our (supervised) Stage 2 models outperform all prior language-specific decompounding tools. Furthermore, a model trained with a Com-poundPiece tokenizer achieves a 5.5% improved performance on compound normalization over an otherwise equivalent SentencePiece model.

**Contributions.** **1)** We introduce a dataset for decompounding of 255k words across 56 languages obtained from Wiktionary. **2)** We show that a byte-level language model can efficiently decompound words via a two-stage training framework, whereas current subword-based LLMs fall short. **3)** We present a way to improve subword tokenization by performing compound segmentation during creation of the tokenizer. **4)** We make our code, models and dataset publicly available at github.com/bminixhofer/compoundpiece.

## 2 Related Work

**Decompounding.** Early work in decompounding used word frequency lists along with manually specified suffixes (e.g., a connective *-s-*) to segment and normalize German compounds (Langer, 1998; Koehn and Knight, 2003). Subsequently, multiple submissions to the Morpho Challenge in morphological segmentation (Kurimo et al., 2010) explicitly or implicitly made use of compound segmentation (Lignos, 2010; Virpioja et al., 2011). Later work replaced the fixed list of suffixes used in Koehn and Knight (2003) by learned morphological operations from parallel corpora (Macherey et al., 2011) or from pre-lemmatized corpora of non-compound words (Ziering and van der Plas, 2016). Another branch of work added more linguistic knowledge in the form of black- and white-lists to the paradigm of Koehn and Knight (2003), resulting in JWordSplitter[2] (German) and nl-splitter[3] (Dutch); this has only been done for a couple of languages due to its knowledge-intensive nature. CharSplit (Tuggener, 2016) achieves high performance for German by relying on the frequency of character n-grams appearing within the compound.

While the approaches above use (at most) light supervision, there exist supervised approaches which learn directly from an annotated corpus of compounds and their constituents, along with optional auxiliary signals (Biemann et al., 2008; Alfonseca et al., 2008). In contrast, SECOS (Riedl and Biemann, 2016) is a fully unsupervised and language-agnostic method achieving competitive performance by using word embeddings along with word frequencies for semantic compound segmen-

---

[2]github.com/danielnaber/jwordsplitter
[3]github.com/bminixhofer/ilps-nl-splitter

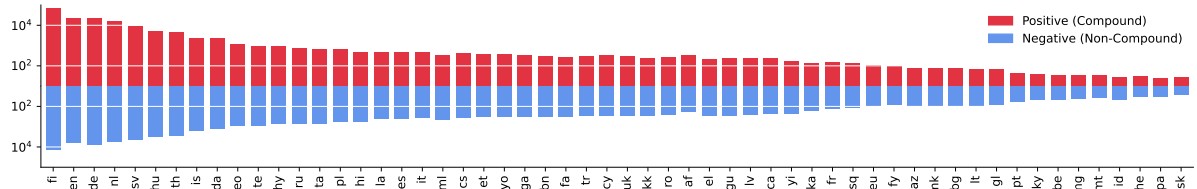

Figure 3: Number of positive and negative examples across languages in the Wiktionary dataset.

| Word | Constituents | Language |
|------|--------------|----------|
| همنیا *(sibling)* | هم *(same)* + نیا *(ancestor)* | Persian |
| akiratis *(horizon)* | akis *(eye)* + ratas *(circle)* | Lithuanian |
| шекара *(border)* | шек *(limit)* + апа *(distance)* | Kazakh |
| Abenteuer *(adventure)* | *None* | German |
| રોકડપ્રવાહ *(cashflow)* | રોકડ *(cash)* + પ્રવાહ *(stream)* | Gujarati |

Figure 4: Example words in the Wiktionary dataset.

tation. Our method improves over SECOS in the unsupervised case and provides a unified alternative to prior language-specific decompounding tools via additional training on labelled data.

**Relation to Morphological Segmentation.** Decompounding can be seen as a special case of morphological segmentation (Batsuren et al., 2022a). However, a large amount of work in morphological segmentation focuses on derivational and inflectional morphology (Cotterell et al., 2016; Faruqui et al., 2016; Cotterell et al., 2018; McCarthy et al., 2019; Goldman et al., 2022), which is reflected by datasets such as UniMorph (Batsuren et al., 2022b) and MorphyNet (Batsuren et al., 2021) annotating inflectional and derivational affixes, but not compound constituents. The SIGMORPHON-2022 Shared Task (Batsuren et al., 2022a, SMST 2022) breaks this pattern by providing a dataset for segmentation into compound constituents in addition to inflectional and derivational affixes. We improve on the SMST 2022 dataset by broadening coverage from 9 to 56 languages, as well as handling negatives (i.e., non-compounds) more carefully (§3.1).

**Decompounding Datasets.** Besides the SMST 2022 dataset, datasets for decompounding include AuCoPro (van Zaanen et al., 2014) for Dutch and Afrikaans, and the GermaNet dataset for German (Henrich and Hinrichs, 2011). Although there is a significant amount of work studying compound terms in languages with highly productive compound formation beyond German and Dutch, such as Finnish and Greek (Pollatsek et al., 2000; Lindén and Pirinen, 2009; Koliopoulou, 2014; Shapiro, 2016; Virkkunen et al., 2018), to the best of our knowledge there exist no public datasets for decompounding in these languages (and beyond).

**Linguistically Informed Tokenization.** Various studies have tried augmenting or replacing the 'linguistically uninformed' subword-tokenizers used in contemporary LMs (Devlin et al., 2019; Raffel et al., 2020, *inter alia*) such as SentencePiece (Kudo and Richardson, 2018) and BPE (Sennrich et al., 2016) with linguistic knowledge. Using manually constructed morphological analyzers before applying BPE (Pan et al., 2020) or after generation (Matthews et al., 2018) has led to improvements, but is limited by the availability (and quality) of morphological analyzers across many languages. Unsupervised morphological segmentation has not shown consistent improvements (Zhou, 2018; Saleva and Lignos, 2021; Domingo et al., 2023); see Mielke et al. (2021) for additional discussion.

## 3 Methodology

### 3.1 Dataset Construction

We use words categorized as compound terms on Wiktionary to create a dataset for decompounding. The information on Wiktionary allows associating compound terms with their corresponding normalized constituents. Since Wiktionary only annotates the top-level split,[4] we recursively split constituents into their smallest parts by checking if the top-level constituents are themselves compound words. Many prior decompounding tools do not evaluate performance on negative examples (i.e. non-compound words; Koehn and Knight, 2003; Riedl and Biemann, 2016; Tuggener, 2016) since most prior datasets do not contain any (Henrich

---

[4]For instance, `highwayman` is segmented into `highway` + `man` instead of `high` + `way` + `man`.

and Hinrichs, 2011; van Zaanen et al., 2014). It is not trivial to obtain negative examples from Wiktionary since a large amount of compound words are not categorized as such, leading to many false negatives. We solve this issue by using all normalized compound constituents as negative examples, since by definition the compound constituents can also appear on their own as non-compound words. Note that this way of obtaining negative examples is biased against words which never occur inside compounds; however, we found this to be a rather weak bias (Appendix E). We include every language with at least 100 words, leading to a dataset which covers 56 languages. The number of training examples is shown in Figure 3, example words in Figure 4. We select up to 1,000 words (but at most 50% of total words) in every language as evaluation data. See Appendix A for further details concerning the dataset.

## 3.2 Two-Stage Training

To overcome the problem of data scarcity in low-resource languages, we introduce a two-stage training procedure for creating dedicated decompounding models. In Stage 1, we train on the *self-supervised objective* of restoring hyphenation in words extracted from a large-scale Web corpus, leading to a self-supervised compound segmentation model. In Stage 2, we fine-tune the model on compounds and their normalized constituents from an annotated corpus in a *supervised fashion*, turning it into a compound normalization model.

**Stage 1: Self-Supervised Compound Segmentation.** This stage is motivated by the fact that hyphen characters can be seen as a *high-precision, low-recall indicator of compound constituent boundaries*, in the same way that newline characters are a high-precision, low-recall indicator of sentence boundaries (Minixhofer et al., 2023). We use this natural segmentation into compound constituents to create a compound segmentation model without requiring any labeled data. First, we obtain all words containing a hyphen plus an equivalent amount of non-hyphenated words from a corpus of unannotated text. Hyphens primarily have two uses: (1) as a compound boundary and (2) to indicate the word continues on the next line. We only want to retain hyphens when they function as compound boundaries, so we filter the instances of (2) by discarding all words where the hyphenated form of the word occurs $x \le e^{-6}$ times less frequent

| | **Word x**: akiratis *(horizon)* | | | |
|---|---|---|---|---|
| | **Norm. constituents c**: {akis *(eye)*, ratas *(circle)*} | | | |

**Find optimal segmentation**

| $s_1$ | $s_2$ | $\mathscr{L}(s_1, c_1)$ | $\mathscr{L}(s_2, c_2)$ | $C(\mathbf{s})$ |
|---|---|---|---|---|
| a | kiratis | 3 | 3 | 6 |
| ak | iratis | 2 | 2 | 4 |
| aki | ratis | 1 | 1 | 2 |
| akir | atis | 1 | 2 | 3 |
| akira | tis | 2 | 3 | 5 |
| akirat | is | 3 | 4 | 7 |
| akirati | s | 4 | 4 | 8 |

**Output**

**Segmentation s⋆**: {aki, ratis}

Figure 5: Turning compound normalization into segmentation by minimizing edit distance (§3.3).

than the non-hyphenated form.[5]

We strip all words of hyphens and train a seq2seq LM to predict the original (hyphenated) form of each word. We introduce a logit bias $b$ added to the logit of the token representing a hyphen to skew generation towards or away from hyphenation at inference time. Training on this data enables effective compound segmentation without relying on human annotations, as demonstrated later in §5.

**Stage 2: Supervised Compound Normalization.** In the second stage, we improve upon the Stage 1 model by additional training on labeled data, where the inputs are individual compounds, and the target is to predict the normalized constituents of each compound, separated by a hyphen. Training exclusively on compound normalization allows using data from the collected Wiktionary dataset, which contains compound terms along with their normalized constituents across many languages, but does not contain compound segmentation annotations.

## 3.3 Turning Normalization into Segmentation

Considering the scarcity of annotated compound segmentation data, it is infeasible to train a multilingual model directly on segmentation. Thus, we introduce a method to predict a segmentation given the normalized constituents. Let $\mathbf{x}$ be a word of length $n$. In addition, we have $k$ normalized com-

---

[5]Consider for example the hyphen-as-compound-boundary in `side-experiments` and the hyphen-as-newline-indicator in `experi-ments`. $\frac{\#\text{experi-ments}}{\#\text{experiments}}$ will be considerably lower than $\frac{\#\text{side-experiments}}{\#\text{sideexperiments}}$. $x$ was chosen from $\{e^{-4}, e^{-5}, e^{-6}, e^{-7}\}$ by manual inspection in preliminary experiments.

pound constituents $c = \{c_1, ..., c_k\}$ (e.g. predicted by the Stage 2 model). Our aim is to find boundaries $r = \{r_0, ..., r_k\}$, $r_0 = 0$, $r_k = n$ giving rise to the segmentation $s = \{x[r_0 : r_1], ..., x[r_{k-1} : r_k]\}$. We approach this problem by minimizing the edit distance of each segment to its corresponding normalized constituent. This leads to an optimization problem where the cost $C(s)$ indicates the total edits needed to turn all segments into their corresponding normalized constituents:

$$C(s) = \sum_{i=1}^{k} \mathcal{L}(s_i, c_i).$$

Here, $\mathcal{L}$ is an edit distance metric such as Levenshtein distance (Levenshtein et al., 1966). The optimal segmentation $s^\star$ is the segmentation with the minimal cost: $s^\star = \arg\min_s C(s)$.

In case of ties, we prefer segmentations with higher edit cost for segments with lower indices due to the preference for languages in our training set for suffixation over prefixation (Hammarström, 2021).[6] There is a total of $\binom{n}{k-1}$ possible segmentations, so solving the optimization problem via enumeration of all solutions is only feasible for short words (Figure 5). We introduce a more efficient search algorithm which is capable of quickly finding the optimal segmentation of long words by enumerating candidates in order of a lower bound on the edit distance in Appendix B. This method can be used to turn the normalization predictions of a model into segmentation. We also use it on the ground-truth normalization from Wiktionary, making it possible to approximate compound segmentation performance by comparing the segmentation corresponding to the ground-truth normalization to the segmentation produced by the model normalization predictions.

### 3.4 Reducing Hard Compounds

We define hard compounds relative to a particular tokenizer as compound words where the constituent boundaries do not coincide with token boundaries set by the tokenizer. More formally, a compound word made up of $k$ constituents and $l$ subwords is hard if the constituent boundaries $r = \{r_0, ..., r_k\}$ are not a subset of the token boundaries $t = \{t_0, ..., t_l\}$ i.e. $r \not\subset t$.

---

[6] E.g., given $x =$ bridesmaid, $c = \{$bride, maid$\}$, we prefer the segmentation $\{$brides, maid$\}$ over $\{$bride, smaid$\}$ although their cost is equal.

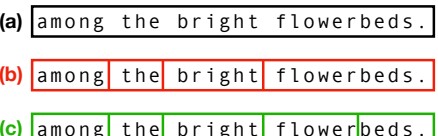

Figure 6: (a) no pretokenization, (b) pretokenization by splitting on whitespace, (c) our pretokenization.

We hypothesize that hard compounds may impair language model performance due to the nontrivial relation of subwords to the compound word. In contrast, in easy compounds the word is naturally decomposed into its constituents. The increased difficulty of hard compounds is apparent on the sequence-to-sequence compound segmentation task: for an easy compound, all tokens can be copied to the output (only the special separator tokens must be inserted). On the other hand, for hard compounds, the tokens change, requiring knowledge of the characters within each token.

Tokenizers where every possible constituent boundary is a token boundary trivially do not give rise to any hard compounds. This includes character-level (Clark et al., 2022; Tay et al., 2022b) as well as byte-level tokenizers (Xue et al., 2022). However, many contemporary language models use subword-based tokenizers to increase efficiency (Devlin et al., 2019; Raffel et al., 2020; Brown et al., 2020). We propose a modification to subword tokenization to reduce the number of hard compounds while keeping the efficiency advantages.

Subword tokenizers typically segment text into pre-tokens (e.g. by splitting on whitespace) before applying their subword tokenization algorithm (Mielke et al., 2021). We propose modifying pretokenization by applying compound segmentation in addition to splitting on whitespace. This modification is only done during creation of the tokenizer, thus incurring no additional cost once the tokenizer has been created. We refer to tokenizers created in this way as *CompoundPiece* tokenizers. The modified pretokenization tries to create more subwords which do not span compound constituent boundaries, thus decreasing the fraction of hard compounds (Figure 6). It aims to turn the dual-route model for computing the meaning of complex (compound) words proposed by Hofmann et al. (2021) into a single-route model which always computes the meaning of compounds from their constituent subwords, and never stores a compound word as a single subword.

## 4 Experimental Setup

### 4.1 Data

We obtain Stage 1 data by selecting all words containing a hyphen from a subset of the mC4 corpus (Xue et al., 2021) which results in ~25M hyphenated words. As negative examples, we choose the $n$ most common words from mC4 such that there is an equivalent amount of non-hyphenated and hyphenated words in every language. Regarding the Stage 2 data, see Section §3.1 before.

### 4.2 Training

We train a decompounding model using a two-stage framework (§3) covering 56 languages. We use ByT5 (Xue et al., 2022) as our main pretrained model and the main starting point since it directly ingests Unicode bytes instead of using subword tokenization, leading to zero hard compounds. We compare our approach against the subword-based T5 (Raffel et al., 2020), Flan-T5 (Chung et al., 2022) and mT5 (Xue et al., 2021) trained with the same two-stage framework. We use t5x (Roberts et al., 2022) for training with a batch size of 512 and a maximum sequence length of 64 tokens, otherwise matching T5 pretraining (Raffel et al., 2020). The setup is the same for Stage 1 and Stage 2.

### 4.3 Evaluation

**Metric.** We measure performance via averaged accuracy, i.e., the ratio of examples which are entirely correctly segmented or normalized.

**Datasets.** Besides our new Wiktionary evaluation subset, we use the established datasets for particular languages: GermaNet (Henrich and Hinrichs, 2011), AuCoPro for Dutch (van Zaanen et al., 2014) as well the subset containing compound-only words across 6 languages from the SIGMORPHON 2022 Shared Task (Batsuren et al., 2022a).[7]

**Baselines.** We use SECOS as the main unsupervised baseline, as well as CharSplit, JWS and nlsplitter as baselines using different amounts of supervision. For the SIGMORPHON 2022 Shared Task dataset, we compare against the task winner, DeepSPIN-3 (Peters and Martins, 2022).

---

[7]We do not include words containing derivational or inflectional affixes since the type of morpheme is not specified, so it is not possible to distinguish between derivational/inflectional affixes and compound constituents. We also do not include root words since we found from manual inspection that >10% of root words are mislabeled, likely due to the difficulty of obtaining negative examples from Wiktionary (§3.1).

**Languages.** For clarity of presentation, we present results on Danish, German, English, Spanish, Estonian, Greek, Persian, Finnish, Hungarian, Kazakh, Latvian, Dutch, Polish and Swedish as a linguistically diverse subset of languages with productive compound formation in the main paper. For the full evaluation across all languages, see Appendix C.

## 5 Results and Discussion

Main compound segmentation results are shown in Table 1. For the self-supervised models, we choose the logit bias $b = 3$ to bias generation towards hyphenated words.[8] ByT5 outperforms subword-based models by a large margin with an absolute 8.9% improvement over the best subword-based model after Stage 1 training, and a 3.7% improvement after Stage 2 training. Comparing models not trained on any annotated data, the self-supervised ByT5 outperforms SECOS on 13 out of 14 languages, and by 13.9% on average.

We further compare against language-specific and supervised methods in Table 2. Our ByT5-based model outperforms all prior methods on every dataset. Since GermaNet tests compound *head* segmentation (i.e., even if a word contains multiple constituents, it is only split into a head and a modifier) we count an example as correctly segmented if either the first constituent matches the modifier or the last constituent matches the head.

**Evaluating LLMs on Decompounding.** We also evaluate in-context learning performance of multiple LLMs on compound segmentation. We use T5 models with 770M, 3B and 11B parameters (Raffel et al., 2020) as well as the UL2 model with 20B parameters (Tay et al., 2022a) since all of them use the same tokenizer, enabling performance comparisons on hard compounds across LLMs. We use the model versions fine-tuned on the Flan dataset collection (Chung et al., 2022), matching our prompt to the style of instructions in the Flan collection (Appendix D). Zero- to 16-shot results are shown in Figure 7. Although the LLMs perform non-trivially well on easy compounds, performance is close to zero (<3%) on hard compounds. Intriguingly, UL2 20B performs worse than Flan T5 XXL (11B), reversing the trend seen on other tasks (Tay et al., 2022a). All the LLMs perform considerably worse than our ByT5-based model; see Figure 1.

---

[8]Chosen among the set {0, 1, 2, 3, 4} to maximize performance on the English validation data.

| | | | da | de | en | es | et | el | fa | fi | hu | kk | lv | nl | pl | sv | *Macro Avg.* |
|---|---|---|---|---|---|---|---|---|---|---|---|---|---|---|---|---|---|
| **P** | | SECOS | 30.0 | 66.5 | 41.2 | 29.0 | 23.4 | 5.3 | 1.4 | 53.1 | 38.8 | 5.0 | 13.9 | 46.8 | 22.2 | 32.2 | 29.2 |
| | S1 | T5 | 55.3 | 56.1 | 85.9 | 69.8 | 29.0 | 0.0 | 0.0 | 31.6 | 48.6 | 16.9 | 29.6 | 44.9 | 36.1 | 53.1 | 39.8 |
| | | Flan-T5 | 58.4 | 58.5 | 89.1 | 71.0 | 37.0 | 0.0 | 0.0 | 33.0 | 53.4 | 17.6 | 41.7 | 44.8 | 40.3 | 56.5 | 42.9 |
| | | mT5 | 25.8 | 38.8 | 79.7 | 58.3 | 18.6 | 21.6 | 3.9 | 24.1 | 18.8 | 45.0 | 20.2 | 23.0 | 32.9 | 21.9 | 30.9 |
| | | ByT5 | 75.6 | 76.0 | 91.3 | 77.2 | 51.6 | 40.9 | 20.9 | 52.7 | 70.0 | 75.9 | 41.7 | 57.2 | 51.8 | 64.8 | 60.5 |
| | S1+S2 | T5 | 86.3 | 96.0 | 95.4 | 82.5 | 77.7 | 0.0 | 0.0 | 98.2 | 89.1 | 18.3 | 69.1 | 94.0 | 78.0 | 89.6 | 69.6 |
| | | Flan-T5 | 86.6 | 95.3 | 95.5 | 83.2 | 80.9 | 0.0 | 0.0 | 98.3 | 87.3 | 16.5 | 68.2 | 93.6 | 77.4 | 89.4 | 69.5 |
| | | mT5 | 87.1 | 94.1 | 95.4 | 82.3 | 83.2 | 73.1 | 62.1 | 97.1 | 90.4 | 86.7 | 76.7 | 93.4 | 84.1 | 90.0 | 85.4 |
| | | ByT5 | **92.2** | **96.6** | **97.8** | **87.1** | **92.6** | **86.1** | **76.6** | **98.8** | **97.2** | **91.7** | **84.8** | **97.5** | **91.2** | **94.3** | **91.7** |
| **N** | | SECOS | 96.1 | 86.6 | 93.8 | 97.4 | **98.6** | 99.7 | 100 | 88.2 | 95.5 | 100 | 100 | 94.1 | 96.9 | 97.3 | 96.0 |
| | S1 | T5 | 88.5 | 91.8 | 91.7 | 88.7 | 82.3 | **100** | **100** | 82.2 | 93.8 | 74.0 | 87.4 | 83.7 | 90.6 | 91.8 | 89.0 |
| | | Flan-T5 | 88.5 | 92.1 | 91.3 | 89.9 | 82.3 | **100** | **100** | 82.9 | 91.6 | 72.9 | 87.0 | 87.0 | 90.4 | 92.4 | 89.2 |
| | | mT5 | 92.7 | 92.8 | 90.9 | 92.3 | 89.9 | 95.3 | 99.3 | 88.2 | 98.0 | 88.0 | 95.9 | 89.1 | 94.5 | 94.8 | 93.0 |
| | | ByT5 | 89.0 | 89.7 | 88.4 | 81.5 | 76.0 | 95.7 | 97.3 | 77.6 | 87.1 | 72.1 | 87.7 | 80.3 | 91.4 | 87.8 | 85.8 |
| | S1+S2 | T5 | 93.3 | 94.5 | 98.3 | 97.8 | 95.1 | **100** | **100** | 95.4 | 99.2 | 91.1 | 97.4 | 97.5 | 98.1 | 96.7 | 96.7 |
| | | Flan-T5 | 94.1 | 95.5 | 97.9 | 95.9 | 95.8 | **100** | **100** | **96.7** | 98.6 | 92.6 | 96.7 | 97.5 | 97.1 | 96.7 | 96.8 |
| | | mT5 | 93.8 | **96.2** | **99.2** | 97.4 | 97.9 | 96.3 | 98.7 | 94.1 | 98.6 | 96.9 | 98.1 | 96.7 | 97.9 | 97.3 | 97.1 |
| | | ByT5 | 95.2 | **96.2** | 98.3 | **98.8** | 97.9 | 97.3 | 97.3 | 95.4 | **99.7** | 99.2 | 98.9 | **97.9** | **99.0** | 97.6 | **97.8** |
| **All** | | SECOS | 53.5 | 72.4 | 53.9 | 63.2 | 56.0 | 60.9 | 52.2 | 58.4 | 59.0 | 50.7 | 61.0 | 58.1 | 57.8 | 53.6 | 57.9 |
| | S1 | T5 | 67.1 | 66.5 | 87.3 | 79.3 | 52.1 | 59.0 | 51.5 | 39.3 | 64.7 | 44.4 | 61.2 | 54.2 | 62.1 | 65.8 | 61.0 |
| | | Flan-T5 | 69.1 | 68.3 | 89.6 | 80.5 | 56.6 | 59.0 | 51.5 | 40.6 | 67.0 | 44.2 | 66.5 | 54.9 | 64.2 | 68.3 | 62.9 |
| | | mT5 | 49.6 | 54.6 | 82.4 | 75.3 | 49.5 | 65.1 | 53.1 | 33.8 | 47.0 | 65.7 | 61.6 | 38.8 | 62.3 | 45.9 | 56.0 |
| | | ByT5 | 80.4 | 80.0 | 90.6 | 79.4 | 62.2 | 73.2 | 60.3 | 56.5 | 76.1 | 74.1 | 66.9 | 62.7 | 70.7 | 72.4 | 71.8 |
| | S1+S2 | T5 | 88.8 | 95.6 | 96.1 | 90.2 | 85.2 | 59.0 | 51.5 | 97.8 | 92.7 | 53.4 | 84.6 | 94.8 | 87.6 | 91.9 | 83.5 |
| | | Flan-T5 | 89.3 | 95.4 | 96.1 | 89.6 | 87.3 | 59.0 | 51.5 | 98.1 | 91.3 | 53.2 | 83.7 | 94.5 | 86.8 | 91.8 | 83.4 |
| | | mT5 | 89.5 | 94.7 | 96.3 | 89.8 | 89.6 | 86.8 | 80.9 | 96.6 | 93.3 | 91.6 | 88.4 | 94.2 | 90.7 | 92.4 | 91.1 |
| | | ByT5 | **93.3** | **96.5** | **97.9** | **92.9** | **94.9** | **92.7** | **87.3** | **98.3** | **98.1** | **95.3** | **92.5** | **97.6** | **94.9** | **95.4** | **94.8** |

Table 1: Accuracy on compounds *(Positives=P)*, non-compound words *(Negatives=N)* and across all examples. We report scores of SECOS as baseline, as well as Stage 1 training only (S1) and Stage 1 plus Stage 2 training (S1+S2).

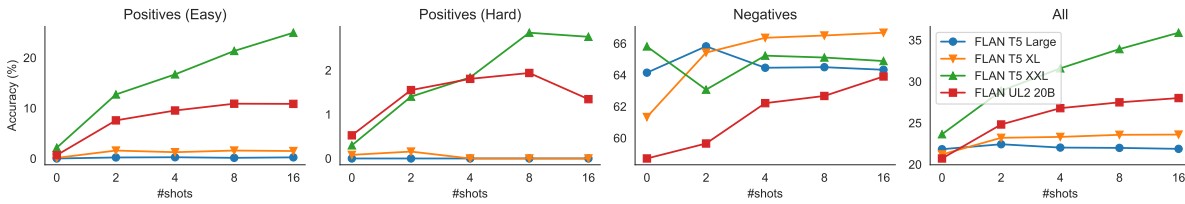

Figure 7: Few-shot in-context learning performance of LLMs on easy positives, hard positives, negatives and across all examples. Hard negatives are the same across all LLMs since they use the same tokenizer.

**Reducing Hard Compounds via Compound-Piece.** To evaluate our method of reducing the number of hard compounds in subword-based language models (§3.4), we train CompoundPiece models in two configurations: (i) multilingual tokenizers across all 56 languages and (ii) separate monolingual tokenizers for every language. For the multilingual tokenizers, we sample languages with $p(L) \propto |L|^{\alpha}$ where $p(L)$ is the probability of sampling text from a language $L$ with $|L|$ texts as in prior work (Conneau et al., 2020). We use a subsample of 10M texts from the mC4 corpus (Xue et al., 2021) with $\alpha = 0.2$. The vocabulary size is 250k for the multilingual and 32k for the monolin-

gual tokenizers, following prior work (Rust et al., 2021; Conneau et al., 2020).

We use our fine-tuned ByT5 model for train-time pretokenization into compound constituents and SentencePiece (Kudo and Richardson, 2018) with Unigram LM (Kudo, 2018) as the subword tokenization applied after pretokenization. As a baseline, we train SentencePiece tokenizers with pretokenization into words (split by whitespace) on the same data. Table 3 shows the percentage of hard compounds for every tokenizer. Compound-Piece reduces the number of hard compounds from 27.1% → 9.7% on average in the monolingual case. In the multilingual case, there is a less marked

|  |  | Segmentation | | | Normalization | | |
|---|---|---|---|---|---|---|---|
|  |  | P | N | All | P | N | All |
| GermaNet | JWS | 83.7 | - | 83.7 | 53.4 | - | 53.4 |
|  | CharSplit | 95.1 | - | 95.1 | - | - | - |
|  | SECOS | 83.6 | - | 83.6 | - | - | - |
|  | ByT5 (S1+S2) | **97.9** | - | **97.9** | **79.6** | - | **79.6** |
| Ours (de) | JWS | 59.7 | 97.6 | 70.8 | 43.2 | **97.6** | 59.1 |
|  | CharSplit | 84.7 | 29.5 | 68.6 | - | - | - |
|  | SECOS | 66.5 | 86.6 | 72.4 | - | - | - |
|  | ByT5 (S1+S2) | **96.6** | 96.2 | **96.5** | **89.8** | 96.2 | **91.7** |
| AuCoPro-nl | nl-splitter | 74.5 | - | 74.5 | 67.1 | - | 67.1 |
|  | SECOS | 59.7 | - | 59.7 | - | - | - |
|  | ByT5 (S1+S2) | **91.7** | - | **91.7** | **76.2** | - | **76.2** |
| Ours (nl) | nl-splitter | 61.2 | 96.7 | 69.7 | 47.0 | 91.2 | 57.6 |
|  | SECOS | 46.8 | 94.1 | 58.1 | - | - | - |
|  | ByT5 (S1+S2) | **97.5** | **97.9** | **97.6** | **87.8** | **97.9** | **90.2** |
| SMST 2022 | DeepSpin-3 | 88.6 | - | 88.6 | 87.3 | - | 87.3 |
|  | ByT5 (S1+S2) | **92.5** | - | **92.5** | **88.6** | - | **88.6** |

Table 2: Comparison against supervised and rule-based baseline models. We use the subset of compound-only words from the Sigmorphon Shared Task (SMST) 2022 data which covers 7 languages (Batsuren et al., 2022a).

| Language | Multilingual | | | Monolingual | |
|---|---|---|---|---|---|
|  | SPM (mT5) | SPM | CPM | SPM | CPM |
| Danish | 15.5 | 16.5 | **12.4** | 24.7 | **5.9** |
| German | 9.9 | 10.3 | **8.2** | 14.6 | **1.8** |
| English | 7.5 | 8.2 | **4.6** | 6.8 | **3.7** |
| Spanish | 29.0 | 24.9 | **18.7** | 14.2 | **10.3** |
| Estonian | 25.5 | 29.5 | **15.2** | 35.4 | **7.2** |
| Greek | 39.9 | 33.6 | **23.1** | 28.9 | **14.9** |
| Persian | 38.6 | 46.1 | **37.2** | 70.9 | **41.8** |
| Finnish | 25.1 | 25.1 | **20.3** | 10.3 | **5.1** |
| Hungarian | 13.8 | 17.1 | **10.1** | 26.1 | **3.7** |
| Kazakh | 14.4 | 13.7 | **9.0** | 28.4 | **4.0** |
| Latvian | 20.2 | 23.8 | **16.1** | 47.5 | **11.7** |
| Dutch | 12.8 | 15.4 | **10.2** | 17.2 | **3.3** |
| Polish | 45.7 | 42.5 | **33.1** | 33.6 | **17.0** |
| Swedish | 13.9 | 17.7 | **12.5** | 21.3 | **5.4** |
| *Macro Avg.* | 22.3 | 23.2 | **16.5** | 27.1 | **9.7** |

Table 3: Percentage of hard compounds after segmentation with different tokenizers. SPM (mT5) is the SentencePiece tokenizer used by mT5 (Xue et al., 2021). SentencePiece (SPM) and CompoundPiece (CPM) tokenizers are trained on text in all 56 languages (Multilingual) and for every language separately (Monolingual).

improvement of 23.2% → 16.5%. This may be because tokens from different languages interfere with the segmentation of any given word. We test this hypothesis by computing plausible token origins for tokens in the multilingual tokenizer. This is done by checking which monolingual tokenizers also contain the token in their vocabulary, and ordering the result by unigram token probability. Examples are shown in Table 4. Interference from

| Word | Monolingual Segmentation | Multilingual Segmentation | Token Origin |
|---|---|---|---|
| tugboat | `_tug`, `boat` | `_tu`, `gbo`, `at` | `_tu`: es, sk, it `gbo`: yo, mg, fr `at`: id, hu, la |
| mindstate | `_mind`, `state` | `_mindst`, `ate` | `_mindst`: da `ate`: it, et, en |
| coatrack | `_coat`, `rack` | `_coa`, `track` | `_coa`: gl, ro `track`: hu, th, da |

Table 4: Example compound words which are easy for the monolingual but hard for the multilingual CompoundPiece tokenizer. "_" indicates whitespace.

| Language | Segmentation | | Normalization | |
|---|---|---|---|---|
|  | SPM-T5 | CPM-T5 | SPM-T5 | CPM-T5 |
| Danish | **77.8** | 77.7 | 65.5 | **69.1** |
| German | **81.0** | 80.7 | 61.5 | **63.8** |
| English | 84.9 | **85.8** | 82.9 | **84.0** |
| Spanish | **75.2** | 74.7 | 50.1 | **55.2** |
| Estonian | 78.6 | **84.5** | 55.1 | **61.3** |
| Greek | **70.6** | 70.0 | 47.1 | **57.8** |
| Persian | 58.2 | **61.2** | 46.6 | **58.1** |
| Finnish | 72.8 | **74.1** | 59.0 | **59.6** |
| Hungarian | 76.2 | **76.9** | 73.3 | **76.2** |
| Kazakh | 72.9 | **75.7** | 59.0 | **74.4** |
| Latvian | **75.2** | 69.1 | 53.5 | **57.3** |
| Dutch | 78.2 | **80.7** | 60.9 | **64.9** |
| Polish | **65.8** | 65.6 | 42.6 | **46.7** |
| Swedish | 76.2 | **77.3** | 61.0 | **65.6** |
| *Macro Avg.* | 74.6 | **75.3** | 58.4 | **63.9** |

Table 5: Accuracy of our multilingual T5 models trained with SentencePiece (SPM-T5) and CompoundPiece (CPM-T5) on segmentation and normalization.

|  | Segmentation | | | Normalization | | |
|---|---|---|---|---|---|---|
|  | P | N | All | P | N | All |
| ByT5 (S1) | 50.8 | **82.5** | **66.6** | 28.5 | **82.5** | **55.2** |
| - hyphen filtering | **53.8** | 62.3 | 58.9 | **30.3** | 62.3 | 47.0 |
| ByT5 (S1+S2) | 80.9 | 98.0 | 89.8 | 58.2 | 97.8 | 78.5 |
| - S1 | 79.3 | 97.3 | 88.6 | 56.8 | 97.1 | 77.4 |

Table 6: Ablation studies on not filtering hyphens-as-newline-indicator and on skipping Stage 1 training.

common tokens in other languages is likely the lead cause for the increased number of hard compounds in the multilingual tokenizers. It could potentially be solved by adjusting token probability based on the input language; we leave this to future work.

To more thoroughly evaluate our tokenization, we train multilingual T5 models using Sentence-Piece and CompoundPiece. We use the same sampling ratio ($\alpha = 0.2$) of mC4 as for creating the tokenizer, but instead use a subset of 500M texts. We match the architecture and the pretraining setup of the mT5-base model, but train for a total of

~65.5B tokens.[9] We evaluate the model on the de-compounding task. Results are shown in Table 5.

**Ablation Studies.** We quantify the impact of the most significant design choices of our model in Table 6. Although filtering hyphens-as-newline-indicator (§4.1) removes only ~300k words (<1%) from the pretraining data, it increases performance on negatives by a large margin. Removing Stage 1 training (i.e., fine-tuning directly on the Wiktionary data instead) consistently decreases performance.

## 6 Conclusion

We systematically investigated word decompounding tasks of compound segmentation and normalization on a wide scale and in multilingual contexts. To this end, we introduced a dataset of 255k words including compounds and non-compounds across 56 languages from Wiktionary, which allowed us to evaluate performance of LLMs on decompounding. We found that current LLMs' performance is limited due to hard compounds which arise when subword token boundaries do not coincide with compound constituent boundaries. We then introduced dedicated models for decompounding which use byte-level tokenization to entirely avoid hard compounds. Finally, we used our decompounding models to create novel CompoundPiece tokenizers, keeping the efficiency advantages of subword tokenization while strongly decreasing the amount of hard compounds; this increases the performance of CompoundPiece models over comparable SentencePiece models on the decompounding tasks.

## Limitations

Although self-supervised training in Stage 1 allows for decompounding without any annotated training data, Stage 2 training is limited to languages with sufficient entries in Wiktionary: this excludes extremely low-resource languages. Furthermore, due to computational constraints we have not trained larger models using CompoundPiece tokenization; hence we are unable to report on its benefits at larger scales and on tasks besides decompounding.

## Acknowledgements

Ivan Vulić is supported by a personal Royal Society University Research Fellowship 'Inclusive and Sustainable Language Technology for a Truly Multilingual World' (no 221137; 2022–).

Research supported with Cloud TPUs from Google's TPU Research Cloud (TRC).

We thank Sebastian Ruder and Srini Narayanan for helpful feedback on a draft of this paper.

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

## A  Dataset Statistics

Statistics for the training and validation splits of the Wiktionary dataset are shown in Table 7.

## B  Efficient Segmentation Algorithm

Pseudocode of the brute-force algorithm to turn normalization into segmentation is shown in Algorithm 1. Since enumerating all possible segmentations is only feasible for short words (§3.3) we introduce a more efficient algorithm (Algorithm 2) where candidate segmentations are ordered such that segmentations with constituents closest in length to the corresponding normalized constituents appear first. Assuming insertions and deletions both have a cost of one (as is the case in standard Levenshtein distance), constituents are thus sorted in increasing order of a lower bound on edit distance. The procedure can stop once the lower bound on edit distance reaches the cost of the best solution found so far since by that point it is impossible for a better solution to be found.

Note that the normalization-to-segmentation problem is related to sequence partitioning (Manne and Sorevik, 1995; Han et al., 1992) where the aim is to find a partition of a sequence such that the maximum cost across partitions of some cost function is minimized. However, since our goal is to find the partitioning with the minimum *aggregated* cost, algorithms for conventional sequence partitioning are not applicable.

## C  Results for All Languages

Segmentation accuracy for all languages is shown in Tables 8-11.

## D  LLM Prompts

The prompt used for LLM evaluations (§5) is shown in Figure 8. The prompt was chosen among 10 prompts to maximize performance on Flan T5 Large. For 2- to 16-shot results, we provide 50% positive (compound) and 50% negative (non-compound) examples in a random order.

## E  Quantifying Negative Collection Bias

We conduct an experiment to measure the extent of the bias against words which do not occur inside compounds in our data collection methodology (§3.1). In particular, we quantify the bias against *long non-compound words*, which usually would not occur inside compounds. We took a

**Zero-shot:**

```
{word}

Hyphenate the above word.
Ans :
```

**n-shot:**

```
{example_0}

Hyphenate the above word.
Ans : {example_0_hyphenated}

...

{example_n}

Hyphenate the above word.
Ans : {example_n_hyphenated}

{word}

Hyphenate the above word.
Ans :
```

Figure 8: Prompts used to evaluate LLM in-context learning compound segmentation performance.

random sample of 500 words each from word frequency lists in English and German (Speer, 2022), manually removed compound words, and compared the length statistics of this (unbiased) sample of non-compounds to our non-compound dataset.

While words in our non-compound dataset are indeed shorter on average (6.0 vs. 6.7 chars for English, 6.7 vs. 7.1 chars for German), with less than one character length difference on average, there is only a weak length bias in data collection.

We also found qualitatively that our non-compound dataset contains a wide variety of words since compounding is typically a process that can occur for many different root words.

**Data:** Compound $x$, norm. constituents $c$.
**Result:** Optimal segmentation $s^\star$.
$k \leftarrow \|c\|, n \leftarrow \|x\|$
$r_0 \leftarrow 0, r_n \leftarrow n$
$\text{best\_cost} \leftarrow \infty$

**for** $r_1, ..., r_{n-1} \in \binom{[n]}{k-1}$ **do**
    Compute $s, C(s)$    /* see §3.3 */
    **if** $C(s) < \text{best\_cost}$ **then**
        $s^{\text{best}} \leftarrow s$
        $\text{best\_cost} \leftarrow C(s)$
    **end**
**end**

$s^\star \leftarrow s^{\text{best}}$

**Algorithm 1:** Naïve brute-force segmentation.

**Data:** Compound $x$, norm. constituents $c$.
**Result:** Optimal segmentation $s^\star$.
$k \leftarrow |c|, n \leftarrow |x|$
$r_0 \leftarrow 0, r_k \leftarrow n$
$\text{best\_cost} \leftarrow \infty$

```
/* Δ is the total difference in
   length of the normalized
   constituents to the word.    */
```
$\Delta = n - \sum_i |c_i|$
$\text{lower\_bound} \leftarrow |\Delta|$

**while** $\text{lower\_bound} < \text{best\_cost}$ **do**
    $\text{offsets} = \{x \mid |x| = k,$
        $\sum_i |x_i| = \text{lower\_bound},$
        $\sum_i x_i = \Delta\}$
    $\text{lower\_bound} \leftarrow \text{lower\_bound} + 1$
    **for** $o_1, ..., o_k \in \text{offsets}$ **do**
        $r_1, ..., r_{k-1} =$
        $|c_1| + o_1, ..., \sum_{i=1}^{n-1} |c_i| + o_i$
        Compute $s, C(s)$ /* see §3.3 */
        **if** $C(s) < \text{best\_cost}$ **then**
            $s^{\text{best}} \leftarrow s$
            $\text{best\_cost} \leftarrow C(s)$
        **end**
    **end**
**end**

$s^\star \leftarrow s^{\text{best}}$

**Algorithm 2:** Segmentation by enumerating candidates in order of increased lower bound on edit distance.

| Language | iso | Training | | | Validation | | |
| | | #Positive | #Negative | Total | #Positive | #Negative | Total |
|---|---|---|---|---|---|---|---|
| Afrikaans | af | 326 | 193 | 519 | 322 | 197 | 519 |
| Azerbaijani | az | 78 | 97 | 175 | 85 | 89 | 174 |
| Belarusian | be | 32 | 47 | 79 | 40 | 38 | 78 |
| Bulgarian | bg | 71 | 89 | 160 | 68 | 92 | 160 |
| Bengali | bn | 301 | 334 | 635 | 304 | 331 | 635 |
| Catalan | ca | 220 | 218 | 438 | 219 | 218 | 437 |
| Czech | cs | 388 | 358 | 746 | 392 | 354 | 746 |
| Welsh | cy | 308 | 273 | 581 | 299 | 281 | 580 |
| Danish | da | 2145 | 1298 | 3443 | 644 | 356 | 1000 |
| German | de | 20743 | 7846 | 28589 | 708 | 292 | 1000 |
| Greek | el | 216 | 292 | 508 | 208 | 299 | 507 |
| English | en | 22896 | 6480 | 29376 | 759 | 241 | 1000 |
| Esperanto | eo | 1097 | 849 | 1946 | 559 | 441 | 1000 |
| Spanish | es | 433 | 401 | 834 | 417 | 417 | 834 |
| Estonian | et | 349 | 315 | 664 | 376 | 288 | 664 |
| Basque | eu | 102 | 98 | 200 | 98 | 101 | 199 |
| Persian | fa | 268 | 314 | 582 | 282 | 300 | 582 |
| Finnish | fi | 69948 | 13314 | 83262 | 848 | 152 | 1000 |
| French | fr | 149 | 135 | 284 | 135 | 148 | 283 |
| Western Frisian | fy | 92 | 85 | 177 | 90 | 86 | 176 |
| Irish | ga | 332 | 322 | 654 | 328 | 325 | 653 |
| Galician | gl | 70 | 79 | 149 | 80 | 69 | 149 |
| Gujarati | gu | 227 | 279 | 506 | 221 | 285 | 506 |
| Hebrew | he | 29 | 34 | 63 | 18 | 44 | 62 |
| Hindi | hi | 472 | 569 | 1041 | 478 | 522 | 1000 |
| Hungarian | hu | 5238 | 3162 | 8400 | 644 | 356 | 1000 |
| Armenian | hy | 872 | 745 | 1617 | 509 | 491 | 1000 |
| Indonesian | id | 26 | 45 | 71 | 32 | 38 | 70 |
| Icelandic | is | 2333 | 1603 | 3936 | 592 | 408 | 1000 |
| Italian | it | 452 | 352 | 804 | 437 | 366 | 803 |
| Georgian | ka | 137 | 156 | 293 | 149 | 143 | 292 |
| Kazakh | kk | 244 | 292 | 536 | 278 | 258 | 536 |
| Kirghiz | ky | 39 | 45 | 84 | 39 | 44 | 83 |
| Latin | la | 450 | 410 | 860 | 452 | 407 | 859 |
| Lithuanian | lt | 65 | 94 | 159 | 76 | 83 | 159 |
| Latvian | lv | 244 | 249 | 493 | 223 | 269 | 492 |
| Malagasy | mg | 35 | 42 | 77 | 32 | 45 | 77 |
| Macedonian | mk | 75 | 94 | 169 | 79 | 90 | 169 |
| Malayalam | ml | 318 | 435 | 753 | 331 | 421 | 752 |
| Maltese | mt | 35 | 36 | 71 | 36 | 35 | 71 |
| Dutch | nl | 15184 | 5258 | 20442 | 761 | 239 | 1000 |
| Panjabi | pa | 24 | 34 | 58 | 19 | 39 | 58 |
| Polish | pl | 628 | 556 | 1184 | 523 | 477 | 1000 |
| Portuguese | pt | 40 | 57 | 97 | 53 | 44 | 97 |
| Romanian | ro | 272 | 261 | 533 | 268 | 265 | 533 |
| Russian | ru | 753 | 718 | 1471 | 507 | 493 | 1000 |
| Slovak | sk | 26 | 28 | 54 | 25 | 29 | 54 |
| Albanian | sq | 124 | 113 | 237 | 109 | 127 | 236 |
| Swedish | sv | 8883 | 4172 | 13055 | 671 | 329 | 1000 |
| Tamil | ta | 656 | 710 | 1366 | 484 | 516 | 1000 |
| Telugu | te | 894 | 909 | 1803 | 507 | 493 | 1000 |
| Thai | th | 4287 | 2754 | 7041 | 614 | 386 | 1000 |
| Turkish | tr | 295 | 287 | 582 | 310 | 271 | 581 |
| Ukrainian | uk | 281 | 291 | 572 | 277 | 295 | 572 |
| Yiddish | yi | 162 | 218 | 380 | 176 | 203 | 379 |
| Yoruba | yo | 349 | 312 | 661 | 348 | 312 | 660 |
| **Total** | | 164713 | 58757 | 223470 | 17539 | 13938 | 31477 |

Table 7: Statistics of the Wiktionary dataset.

| | | | af | az | be | bg | bn | ca | cs | cy | da | de | el | en | eo | es | *Macro Avg.* |
|---|---|---|---|---|---|---|---|---|---|---|---|---|---|---|---|---|---|
| **N** | | SECOS | - | - | - | 7.4 | - | 4.1 | 20.2 | - | 30.0 | 66.5 | 5.3 | 41.2 | - | 29.0 | - |
| | S1 | T5 | 47.5 | 64.7 | 20.0 | 14.7 | 0.0 | 61.2 | 30.6 | 41.1 | 55.3 | 56.1 | 0.0 | 85.9 | 65.3 | 69.8 | 43.7 |
| | | FLAN T5 | 52.8 | 69.4 | 17.5 | 16.2 | 0.0 | 59.8 | 36.0 | 43.8 | 58.4 | 58.5 | 0.0 | 89.1 | 67.6 | 71.0 | 45.7 |
| | | mT5 | 22.7 | 34.1 | 20.0 | 10.3 | 14.5 | 50.7 | 28.8 | 36.5 | 25.8 | 38.8 | 21.6 | 79.7 | 44.0 | 58.3 | 34.7 |
| | | ByT5 | 64.9 | 70.6 | 45.0 | 29.4 | 34.5 | 68.5 | 48.2 | 49.8 | 75.6 | 76.0 | 40.9 | 91.3 | 82.3 | 77.2 | 61.0 |
| | S1+S2 | T5 | 83.9 | 75.3 | 22.5 | 35.3 | 0.0 | 70.8 | 68.9 | 60.2 | 86.3 | 96.0 | 0.0 | 95.4 | 78.7 | 82.5 | 61.1 |
| | | FLAN T5 | 84.8 | 74.1 | 22.5 | 33.8 | 0.0 | 75.3 | 67.3 | 59.5 | 86.6 | 95.3 | 0.0 | 95.5 | 77.6 | 83.2 | 61.1 |
| | | mT5 | 83.5 | 85.9 | 70.0 | 76.5 | 79.6 | 71.7 | 76.8 | 57.9 | 87.1 | 94.1 | 73.1 | 95.4 | 78.2 | 82.3 | 79.4 |
| | | ByT5 | **90.4** | **89.4** | **80.0** | **79.4** | **91.1** | **81.7** | **85.5** | **72.9** | **92.2** | **96.6** | **86.1** | **97.8** | **89.8** | **87.1** | **87.1** |
| **P** | | SECOS | - | - | - | **100** | - | **99.1** | **100** | - | **96.1** | 86.6 | 99.7 | 93.8 | - | 97.4 | - |
| | S1 | T5 | 87.8 | 84.3 | 63.2 | 73.9 | **100** | 88.1 | 80.5 | 74.4 | 88.5 | 91.8 | **100** | 91.7 | 83.9 | 88.7 | 85.5 |
| | | FLAN T5 | 90.4 | 85.4 | 71.1 | 73.9 | **100** | 92.2 | 79.9 | 74.4 | 88.5 | 92.1 | **100** | 91.3 | 86.2 | 89.9 | 86.8 |
| | | mT5 | **95.4** | 91.0 | 81.6 | 93.5 | 95.8 | 93.6 | 96.3 | 82.9 | 92.7 | 92.8 | 95.3 | 90.9 | 90.2 | 92.3 | 91.7 |
| | | ByT5 | 87.8 | 86.5 | 57.9 | 72.8 | 93.1 | 84.9 | 87.9 | 70.5 | 89.0 | 89.7 | 95.7 | 88.4 | 76.6 | 81.5 | 83.0 |
| | S1+S2 | T5 | 91.9 | 97.8 | 94.7 | 96.7 | **100** | 97.2 | 94.6 | 93.6 | 93.3 | 94.5 | **100** | 98.3 | 97.7 | 97.8 | 96.3 |
| | | FLAN T5 | 90.9 | 96.6 | 94.7 | 95.7 | **100** | 95.4 | 93.5 | **96.1** | 94.1 | 95.5 | **100** | 97.9 | 97.3 | 95.9 | 96.0 |
| | | mT5 | 92.4 | 98.0 | **100** | **100** | 97.0 | 95.9 | 97.7 | 95.7 | 93.8 | **96.2** | 96.3 | **99.2** | **98.0** | 97.4 | 97.1 |
| | | ByT5 | 93.4 | 98.9 | **100** | **100** | 97.3 | 97.2 | 98.0 | 95.7 | 95.2 | **96.2** | 97.3 | 98.3 | **98.0** | 98.8 | **97.5** |
| **All** | | SECOS | - | - | - | 60.6 | - | 51.5 | 58.0 | - | 53.5 | 72.4 | 60.9 | 53.9 | - | 63.2 | - |
| | S1 | T5 | 62.8 | 74.7 | 41.0 | 48.8 | 52.1 | 74.6 | 54.3 | 57.2 | 67.1 | 66.5 | 59.0 | 87.3 | 73.5 | 79.3 | 64.2 |
| | | FLAN T5 | 67.1 | 77.6 | 43.6 | 49.4 | 52.1 | 76.0 | 56.8 | 58.6 | 69.1 | 68.3 | 59.0 | 89.6 | 75.8 | 80.5 | 66.0 |
| | | mT5 | 50.3 | 63.2 | 50.0 | 58.1 | 56.9 | 72.1 | 60.9 | 59.0 | 49.6 | 54.6 | 65.1 | 82.4 | 64.4 | 75.3 | 61.6 |
| | | ByT5 | 73.6 | 78.7 | 51.3 | 54.4 | 65.0 | 76.7 | 67.0 | 59.8 | 80.4 | 80.0 | 73.2 | 90.6 | 79.8 | 79.4 | 72.1 |
| | S1+S2 | T5 | 86.9 | 86.8 | 57.7 | 70.6 | 52.1 | 84.0 | 81.1 | 76.4 | 88.8 | 95.6 | 59.0 | 96.1 | 87.1 | 90.2 | 79.5 |
| | | FLAN T5 | 87.1 | 85.6 | 57.7 | 69.4 | 52.1 | 85.4 | 79.8 | 77.2 | 89.3 | 95.4 | 59.0 | 96.1 | 86.3 | 89.6 | 79.3 |
| | | mT5 | 86.9 | 93.1 | 84.6 | 90.0 | 88.7 | 83.8 | 86.7 | 76.2 | 89.5 | 94.7 | 86.8 | 96.3 | 86.9 | 89.8 | 88.1 |
| | | ByT5 | **91.5** | **94.3** | **89.7** | **91.2** | **94.3** | **89.5** | **91.4** | **84.0** | **93.3** | **96.5** | **92.7** | **97.9** | **93.4** | **92.9** | **92.3** |

Table 8: Accuracy on languages af-es.

| | | | et | eu | fa | fi | fr | fy | ga | gl | gu | he | hi | hu | hy | id | *Macro Avg.* |
|---|---|---|---|---|---|---|---|---|---|---|---|---|---|---|---|---|---|
| **N** | | SECOS | 23.4 | 4.1 | 1.4 | 53.1 | 11.9 | - | - | 2.5 | - | - | - | 38.8 | - | - | - |
| | S1 | T5 | 29.0 | 28.6 | 0.0 | 31.6 | 31.9 | 53.3 | 69.8 | 50.0 | 0.0 | 0.0 | 0.0 | 48.6 | 0.0 | 34.4 | 26.9 |
| | | FLAN T5 | 37.0 | 31.6 | 0.0 | 33.0 | 31.9 | 58.9 | 70.1 | 51.2 | 0.0 | 0.0 | 0.0 | 53.4 | 0.0 | 40.6 | 29.1 |
| | | mT5 | 18.6 | 18.4 | 3.9 | 24.1 | 21.5 | 24.4 | 59.8 | 38.8 | 59.3 | 22.2 | 39.7 | 18.8 | 4.3 | 12.5 | 26.2 |
| | | ByT5 | 51.6 | 42.9 | 20.9 | 52.7 | 44.4 | 52.2 | 76.8 | 52.5 | 79.6 | 38.9 | 66.5 | 70.0 | 10.2 | 50.0 | 50.7 |
| | S1+S2 | T5 | 77.7 | 38.8 | 0.0 | 98.2 | 48.1 | 84.4 | 83.2 | 65.0 | 0.0 | 0.0 | 0.0 | 89.1 | 0.0 | 46.9 | 45.1 |
| | | FLAN T5 | 80.9 | 41.8 | 0.0 | 98.3 | 49.6 | 86.7 | 81.4 | 60.0 | 0.0 | 0.0 | 0.0 | 87.3 | 0.0 | 46.9 | 45.2 |
| | | mT5 | 83.2 | 50.0 | 62.1 | 97.1 | 48.9 | 81.1 | 82.6 | 62.5 | 85.1 | **44.4** | 81.8 | 90.4 | 77.2 | 40.6 | 70.5 |
| | | ByT5 | **92.6** | **58.2** | **76.6** | **98.8** | **62.2** | **91.1** | **88.7** | **67.5** | **90.0** | 33.3 | **88.9** | **97.2** | **85.1** | **53.1** | **77.4** |
| **P** | | SECOS | **98.6** | **100** | **100** | 88.2 | 97.3 | - | - | 95.7 | - | - | - | 95.5 | - | - | - |
| | S1 | T5 | 82.3 | 85.1 | **100** | 82.2 | 94.6 | 87.2 | 82.2 | 82.6 | **100** | **100** | **100** | 93.8 | **100** | 81.6 | 90.8 |
| | | FLAN T5 | 82.3 | 87.1 | **100** | 82.9 | 98.0 | 87.2 | 76.0 | 94.2 | **100** | **100** | **100** | 91.6 | **100** | 76.3 | 91.1 |
| | | mT5 | 89.9 | 89.1 | 99.3 | 88.2 | 95.3 | 90.7 | 88.3 | 95.7 | 97.9 | 97.7 | 99.4 | 98.0 | 97.6 | 78.9 | 93.3 |
| | | ByT5 | 76.0 | 71.3 | 97.3 | 77.6 | 91.9 | 88.4 | 77.8 | 71.0 | 94.7 | 95.5 | 98.3 | 87.1 | 92.5 | 57.9 | 84.1 |
| | S1+S2 | T5 | 95.1 | 98.0 | **100** | 95.4 | 97.3 | **100** | 98.8 | 95.7 | **100** | **100** | **100** | 99.2 | **100** | **100** | 98.5 |
| | | FLAN T5 | 95.8 | 96.0 | **100** | **96.7** | 97.3 | **100** | 97.5 | 95.7 | **100** | **100** | **100** | 98.6 | **100** | **100** | 98.4 |
| | | mT5 | 97.9 | 97.0 | 98.7 | 94.1 | **98.6** | 97.7 | 98.2 | **100** | 97.2 | 97.7 | 99.0 | 98.6 | 97.4 | **100** | 98.0 |
| | | ByT5 | 97.9 | 97.0 | 97.3 | 95.4 | **98.6** | **100** | 98.8 | **100** | 97.9 | **100** | 99.2 | **99.7** | 98.2 | **100** | **98.6** |
| **All** | | SECOS | 56.0 | 52.8 | 52.2 | 58.4 | 56.5 | - | - | 45.6 | - | - | - | 59.0 | - | - | - |
| | S1 | T5 | 52.1 | 57.3 | 51.5 | 39.3 | 64.7 | 69.9 | 76.0 | 65.1 | 56.3 | 71.0 | 52.2 | 64.7 | 49.1 | 60.0 | 59.2 |
| | | FLAN T5 | 56.6 | 59.8 | 51.5 | 40.6 | 66.4 | 72.7 | 73.0 | 71.1 | 56.3 | 71.0 | 52.2 | 67.0 | 49.1 | 60.0 | 60.5 |
| | | mT5 | 49.5 | 54.3 | 53.1 | 33.8 | 60.1 | 56.8 | 74.0 | 65.1 | 81.0 | 75.8 | 70.9 | 47.0 | 50.1 | 48.6 | 58.6 |
| | | ByT5 | 62.2 | 57.3 | 60.3 | 56.5 | 69.3 | 69.9 | 77.3 | 61.1 | 88.1 | 79.0 | 83.1 | 76.1 | 50.6 | 54.3 | 67.5 |
| | S1+S2 | T5 | 85.2 | 68.8 | 51.5 | 97.8 | 73.9 | 92.0 | 91.0 | 79.2 | 56.3 | 71.0 | 52.2 | 92.7 | 49.1 | 75.7 | 74.0 |
| | | FLAN T5 | 87.3 | 69.3 | 51.5 | 98.1 | 74.6 | 93.2 | 89.4 | 76.5 | 56.3 | 71.0 | 52.2 | 91.3 | 49.1 | 75.7 | 74.0 |
| | | mT5 | 89.6 | 73.9 | 80.9 | 96.6 | 74.9 | 89.2 | 90.4 | 79.9 | 91.9 | **82.3** | 90.8 | 93.3 | 87.1 | 72.9 | 85.3 |
| | | ByT5 | **94.9** | **77.9** | **87.3** | **98.3** | **81.3** | **95.5** | **93.7** | **82.6** | **94.5** | 80.6 | **94.3** | **98.1** | **91.5** | **78.6** | **89.2** |

Table 9: Accuracy on languages et-id.

|  |  |  | is | it | ka | kk | ky | la | lt | lv | mg | mk | ml | mt | nl | pa | *Macro Avg.* |
|---|---|---|---|---|---|---|---|---|---|---|---|---|---|---|---|---|---|
| N | | SECOS | - | 32.5 | - | 5.0 | - | 5.3 | - | 13.9 | - | - | - | - | 46.8 | - | - |
| | S1 | T5 | 41.4 | 42.6 | 0.0 | 16.9 | 15.4 | 29.0 | 21.1 | 29.6 | 31.2 | 15.2 | 0.0 | 19.4 | 44.9 | 0.0 | 21.9 |
| | | FLAN T5 | 45.4 | 48.5 | 0.0 | 17.6 | 12.8 | 33.0 | 28.9 | 41.7 | 34.4 | 17.7 | 0.0 | 25.0 | 44.8 | 0.0 | 25.0 |
| | | mT5 | 26.4 | 26.8 | 21.5 | 45.0 | 30.8 | 21.9 | 11.8 | 20.2 | 25.0 | 24.1 | 18.4 | 25.0 | 23.0 | 42.1 | 25.8 |
| | | ByT5 | 65.9 | 56.1 | 61.7 | 75.9 | 64.1 | 33.6 | 25.0 | 41.7 | 37.5 | 36.7 | 33.5 | 27.8 | 57.2 | 68.4 | 48.9 |
| | S1+S2 | T5 | 78.7 | 68.6 | 0.0 | 18.3 | 23.1 | 59.3 | 65.8 | 69.1 | 53.1 | 49.4 | 0.0 | 41.7 | 94.0 | 0.0 | 44.4 |
| | | FLAN T5 | 78.7 | 68.9 | 0.0 | 16.5 | 17.9 | 61.7 | 63.2 | 68.2 | 40.6 | 49.4 | 0.0 | 41.7 | 93.6 | 0.0 | 42.9 |
| | | mT5 | 82.9 | 68.9 | 82.6 | 86.7 | 79.5 | 61.9 | 60.5 | 76.7 | 53.1 | 72.2 | 68.0 | 52.8 | 93.4 | 63.2 | 71.6 |
| | | ByT5 | **90.5** | **81.2** | **83.9** | **91.7** | **84.6** | **73.5** | **80.3** | **84.8** | **65.6** | **88.6** | **83.7** | **58.3** | **97.5** | **78.9** | **81.7** |
| P | | SECOS | - | 97.0 | - | **100** | - | **99.8** | - | **100** | - | - | - | - | 94.1 | - | - |
| | S1 | T5 | 83.1 | 88.8 | 99.3 | 74.0 | 77.3 | 80.3 | 81.9 | 87.4 | 68.9 | 68.9 | **100** | 91.4 | 83.7 | **100** | 84.6 |
| | | FLAN T5 | 80.1 | 91.3 | 99.3 | 72.9 | 86.4 | 82.8 | 86.7 | 87.0 | 75.6 | 66.7 | **100** | 85.7 | 87.0 | **100** | 85.8 |
| | | mT5 | 90.0 | 92.1 | 97.2 | 88.0 | 95.5 | 84.5 | 85.5 | 95.9 | 84.4 | 90.0 | 96.9 | 97.1 | 89.1 | **100** | 91.9 |
| | | ByT5 | 82.1 | 83.6 | 88.8 | 72.1 | 86.4 | 56.0 | 71.1 | 87.7 | 57.8 | 85.6 | 72.7 | 94.3 | 80.3 | **100** | 79.9 |
| | S1+S2 | T5 | 96.1 | 96.4 | **100** | 91.1 | 97.7 | 94.8 | 92.8 | 97.4 | 88.9 | 93.3 | **100** | 94.3 | 97.5 | **100** | 95.7 |
| | | FLAN T5 | 95.6 | 96.2 | 99.3 | 92.6 | 97.7 | 95.6 | 95.2 | 96.7 | **91.1** | 93.3 | **100** | 97.1 | 97.5 | **100** | 96.3 |
| | | mT5 | 95.6 | 97.3 | 98.6 | 96.9 | **100** | 96.1 | **96.4** | 98.1 | 82.2 | 96.7 | 98.3 | 88.6 | 96.7 | **100** | 95.8 |
| | | ByT5 | **97.1** | **97.8** | 99.3 | 99.2 | **100** | 98.5 | 94.0 | 98.9 | **91.1** | 98.9 | 98.8 | **100** | 97.9 | **100** | 98.0 |
| All | | SECOS | - | 61.9 | - | 50.7 | - | 50.1 | - | 61.0 | - | - | - | - | 58.1 | - | - |
| | S1 | T5 | 58.4 | 63.6 | 48.6 | 44.4 | 48.2 | 53.3 | 52.8 | 61.2 | 53.2 | 43.8 | 56.0 | 54.9 | 54.2 | 67.2 | 54.3 |
| | | FLAN T5 | 59.6 | 68.0 | 48.6 | 44.2 | 51.8 | 56.6 | 59.1 | 66.5 | 58.4 | 43.8 | 56.0 | 54.9 | 54.9 | 67.2 | 56.4 |
| | | mT5 | 52.3 | 56.5 | 58.6 | 65.7 | 65.1 | 51.6 | 50.3 | 61.6 | 59.7 | 59.2 | 62.4 | 60.6 | 38.8 | 81.0 | 58.8 |
| | | ByT5 | 72.5 | 68.6 | 75.0 | 74.1 | 75.9 | 44.2 | 49.1 | 66.9 | 49.4 | 62.7 | 55.5 | 60.6 | 62.7 | 89.7 | 64.8 |
| | S1+S2 | T5 | 85.8 | 81.3 | 49.0 | 53.4 | 62.7 | 76.1 | 79.9 | 84.6 | 74.0 | 72.8 | 56.0 | 67.6 | 94.8 | 67.2 | 71.8 |
| | | FLAN T5 | 85.6 | 81.3 | 48.6 | 53.2 | 60.2 | 77.8 | 79.9 | 83.7 | 70.1 | 72.8 | 56.0 | 69.0 | 94.5 | 67.2 | 71.4 |
| | | mT5 | 88.1 | 81.8 | 90.4 | 91.6 | 90.4 | 78.1 | 79.2 | 88.4 | 70.1 | 85.2 | 85.0 | 70.4 | 94.2 | 87.9 | 84.4 |
| | | ByT5 | **93.2** | **88.8** | **91.4** | **95.3** | **92.8** | **85.3** | **87.4** | **92.5** | **80.5** | **94.1** | **92.2** | **78.9** | **97.6** | **93.1** | **90.2** |

Table 10: Accuracy on languages is-pa.

|  |  |  | pl | pt | ro | ru | sk | sq | sv | ta | te | th | tr | uk | yi | yo | *Macro Avg.* |
|---|---|---|---|---|---|---|---|---|---|---|---|---|---|---|---|---|---|
| N | | SECOS | 22.2 | 9.4 | 7.8 | 35.9 | - | - | 32.2 | - | - | - | 7.7 | - | - | - | - |
| | S1 | T5 | 36.1 | 30.2 | 51.9 | 22.3 | 12.0 | 29.4 | 53.1 | 0.0 | 0.0 | 0.0 | 28.1 | 17.7 | 0.0 | 12.9 | 21.0 |
| | | FLAN T5 | 40.3 | 47.2 | 55.6 | 25.6 | 16.0 | 31.2 | 56.5 | 0.0 | 0.0 | 0.0 | 34.8 | 22.0 | 0.0 | 16.1 | 24.7 |
| | | mT5 | 32.9 | 20.8 | 47.8 | 15.8 | 16.0 | 22.9 | 21.9 | 19.6 | 40.2 | 9.3 | 15.2 | 13.0 | 36.4 | 10.9 | 23.0 |
| | | ByT5 | 51.8 | 45.3 | 61.2 | 31.0 | 36.0 | 34.9 | 64.8 | 46.1 | 61.7 | 27.0 | 32.6 | 35.7 | 50.6 | 18.7 | 42.7 |
| | S1+S2 | T5 | 78.0 | 49.1 | 63.8 | 50.3 | 48.0 | 52.3 | 89.6 | 0.0 | 0.0 | 0.0 | 67.4 | 37.5 | 0.0 | 19.0 | 39.6 |
| | | FLAN T5 | 77.4 | 49.1 | 64.9 | 50.1 | 60.0 | 45.9 | 89.4 | 0.0 | 0.0 | 0.0 | 66.1 | 36.1 | 0.0 | 19.0 | 39.9 |
| | | mT5 | 84.1 | 39.6 | 65.7 | 77.9 | 60.0 | 45.0 | 90.0 | 61.0 | 81.9 | 83.7 | 71.6 | 76.9 | 76.1 | 31.3 | 67.5 |
| | | ByT5 | **91.2** | **56.6** | **73.5** | **91.3** | **72.0** | **56.0** | **94.3** | **73.6** | **84.4** | **90.6** | **82.3** | **86.6** | **83.0** | **48.0** | **77.4** |
| P | | SECOS | 96.9 | **97.7** | 95.8 | 92.1 | - | - | 97.3 | - | - | - | **100** | - | - | - | - |
| | S1 | T5 | 90.6 | 86.4 | 90.9 | 66.1 | 89.7 | 80.3 | 91.8 | **100** | **100** | 99.5 | 87.5 | 68.5 | **100** | 86.2 | 88.4 |
| | | FLAN T5 | 90.4 | 88.6 | 91.3 | 67.1 | 86.2 | 81.1 | 92.4 | **100** | **100** | 99.7 | 89.3 | 72.2 | **100** | 85.3 | 88.8 |
| | | mT5 | 94.5 | 81.8 | 94.7 | 83.6 | 96.6 | 96.1 | 94.8 | 94.4 | 98.4 | 97.9 | 94.8 | 87.1 | 99.5 | 92.9 | 93.4 |
| | | ByT5 | 91.4 | 79.5 | 86.8 | 67.3 | 86.2 | 83.5 | 87.8 | 63.2 | 95.7 | 89.6 | 81.2 | 72.2 | 92.1 | 85.6 | 83.0 |
| | S1+S2 | T5 | 98.1 | **97.7** | 95.8 | 95.9 | 96.6 | 93.7 | 96.7 | 99.8 | **100** | **100** | 95.6 | 97.3 | **100** | 93.9 | 97.2 |
| | | FLAN T5 | 97.1 | **97.7** | **97.7** | 95.9 | 96.6 | 95.3 | 96.7 | **100** | **100** | **100** | 96.3 | 98.6 | **100** | 97.1 | 97.8 |
| | | mT5 | 97.9 | **97.7** | 95.8 | 98.6 | **100** | **99.2** | 97.3 | 94.6 | 97.8 | 97.2 | 98.5 | **99.7** | 97.0 | 97.8 | 97.8 |
| | | ByT5 | **99.0** | **97.7** | 96.6 | **99.0** | 96.6 | 97.6 | **97.6** | 96.9 | 98.8 | 99.0 | 98.2 | 99.3 | 98.0 | **98.1** | **98.0** |
| All | | SECOS | 57.8 | 49.5 | 51.6 | 63.6 | - | - | 53.6 | - | - | - | 50.8 | - | - | - | - |
| | S1 | T5 | 62.1 | 55.7 | 71.3 | 43.9 | 53.7 | 56.8 | 65.8 | 51.6 | 49.3 | 38.4 | 55.8 | 43.9 | 53.6 | 47.6 | 53.5 |
| | | FLAN T5 | 64.2 | 66.0 | 73.4 | 46.1 | 53.7 | 58.1 | 68.3 | 51.6 | 49.3 | 38.5 | 60.2 | 47.9 | 53.6 | 48.8 | 55.7 |
| | | mT5 | 62.3 | 48.5 | 71.1 | 49.2 | 59.3 | 62.3 | 45.9 | 58.2 | 68.9 | 43.5 | 52.3 | 51.2 | 70.2 | 49.7 | 56.6 |
| | | ByT5 | 70.7 | 60.8 | 73.9 | 48.9 | 63.0 | 61.0 | 72.4 | 54.9 | 78.5 | 51.2 | 55.2 | 54.5 | 72.8 | 50.3 | 62.0 |
| | S1+S2 | T5 | 87.6 | 71.1 | 79.7 | 72.8 | 74.1 | 74.6 | 91.9 | 51.5 | 49.3 | 38.6 | 80.6 | 68.4 | 53.6 | 54.4 | 67.7 |
| | | FLAN T5 | 86.8 | 71.1 | 81.2 | 72.7 | 79.6 | 72.5 | 91.9 | 51.6 | 49.3 | 38.6 | 80.2 | 68.4 | 53.6 | 55.9 | 68.1 |
| | | mT5 | 90.7 | 66.0 | 80.7 | 88.1 | 81.5 | 74.2 | 92.4 | 78.3 | 89.7 | 88.9 | 84.2 | 88.6 | 87.3 | 62.7 | 82.4 |
| | | ByT5 | **94.9** | **75.3** | **85.0** | **95.1** | **85.2** | **78.4** | **95.4** | **85.6** | **91.5** | **93.8** | **89.7** | **93.2** | **91.0** | **71.7** | **87.5** |

Table 11: Accuracy on languages pl-yo.