# OpenReview forum: "CompoundPiece: Evaluating and Improving Decompounding Performance of Language Models"
_EMNLP/2023/Conference — EMNLP 2023 Main_

### Official Review · Reviewer_8bjC · 2023-08-02

**Soundness:** 3

**Excitement:**

4: Strong: This paper deepens the understanding of some phenomenon or lowers the barriers to an existing research direction.

**Paper Topic And Main Contributions:**

This paper creates a multi-lingual data collection that contains information on compounds and their boundaries. Using the data collection, a number of systems are developed that aim to automatically identify the compound boundaries.

**Questions For The Authors:**

In footnote 5 you highlight the problem of obtaining suitable non-compounds.  This may indeed be a problem as the non-compounds you select (if I understand correctly) also occur as part in the compounds.

I don't understand your point 1 on line 261. In many languages, hyphens are never used to indicate a compound boundary.

On line 386 you mention you use n most common words.  Typically the most common words are shorter than the compounds.  How does this influence the results?


**Reasons To Accept:**

Compounding is an active process in many languages. This introduces difficulties for instance related to dictionary development, spelling checking, etc.
The data collection contains interesting data that can also serve the basis for more linguistically oriented research.

**Reasons To Reject:**

The process of creating the data collection has limitations. In particular, the process of finding non-compounds has a strong bias (if I understand correctly a bias towards short words).  Furthermore, assumptions are made related to the use of hyphens (see below).

**Reproducibility:**

3: Could reproduce the results with some difficulty. The settings of parameters are underspecified or subjectively determined; the training/evaluation data are not widely available.

**Reviewer Confidence:**

4: Quite sure. I tried to check the important points carefully. It's unlikely, though conceivable, that I missed something that should affect my ratings.

---

> ### Author Rebuttal · Authors · 2023-08-28
>
> Thank you for very much your feedback and for bringing up the valid concern about non-compound data collection being strongly biased towards short words.
>
> We ran an additional experiment to quantify if such a bias exists (and to what extent): We took a random sample of 500 words each from word frequency lists in English and German [1], manually removed compound words, and compared the length statistics of this (unbiased) sample of non-compounds to our (biased) non-compound dataset.
>
> While words in our non-compound dataset are indeed shorter (avg. 6.0 vs 6.7 characters for English, avg. 6.7 vs 7.1 characters for German), we would argue that with <1 character length difference on average, there is only a weak length bias in data collection. We will add this result to the paper to mitigate the concern.
>
> We also found qualitatively that our non-compound dataset contains a wide variety of words since compounding is typically a process that can occur for many different root words.
>
> To address your remaining questions:
>
> > I don't understand your point 1 on line 261. In many languages, hyphens are never used to indicate a compound boundary.
>
> We found that hyphens are used to indicate compound boundaries in all 56 languages which we investigated. This may not be formally correct in all languages, but it is present in every one of them in text from the internet.
>
> > On line 386 you mention you use n most common words. Typically the most common words are shorter than the compounds. How does this influence the results?
>
> This is a practical choice purely for Stage 1 to reduce noise. Since n is very large (avg. >400k per language) this does not particularly favour short words. We will clarify this in the paper.
>
> [1]: via https://github.com/rspeer/wordfreq

---

### Official Review · Reviewer_WAo7 · 2023-08-02

**Soundness:** 4

**Excitement:**

4: Strong: This paper deepens the understanding of some phenomenon or lowers the barriers to an existing research direction.

**Paper Topic And Main Contributions:**

Compounding is pervasive in many of the world's languages, but most computational work on analyzing compounds has focused on a handful of languages. This paper studies the task of decompounding, namely analyzing compounds to their constituent parts. The work makes several contributions in this respect: (1) it compiles a benchmark of compounds in 56 diverse languages, by mining them from Wikitionary; (2) it defines two variants of the task of decompounding - one defined as segmentation and the other as extracting the stems; (3) it sets a benchmark in terms of the performance of existing models on this task, and (4) it proposes a new self-supervised method for compound segmentation.


**Questions For The Authors:**

-- Did you spot any interesting cross-linguistic trends in your results? e.g., by language families?

-- Are the developed algorithmic methods relevant for other morphological tasks or other segmentation tasks?

**Reasons To Accept:**

-- The paper addresses an important task, that is or should be part of the basic analysis pipeline of many languages.
-- The paper extends the scope of the task to a diverse set of languages, which is especially commendable given the limited scope of languages explored in previous work. The resulting dataset is large: 255K compound words and I am sure it will be useful in future work.
-- The paper proposes a nice algorithmic method for self-supervised learning of segmentation, using a nice trick based on the use of hyphens.
-- Sound experimental setup, and good results over an extensive set of settings.
-- The paper is well structured and well presented.

**Reasons To Reject:**

I don't see any major risks in publishing it. One possible minor risk is that the method presented in the work is that I do not think it will likely to make a wide impact outside the scope of the task it addresses. The paper could have potentially made a contribution to the linguistic literature but does not include much discussion of the linguistic implications of its results.
The same goes for technical contribution: while the method appears sound and effective, little discussion is made as to whether the effectiveness of the method is likely to make an impact for other tasks or sub-fields.

**Reproducibility:**

4: Could mostly reproduce the results, but there may be some variation because of sample variance or minor variations in their interpretation of the protocol or method.

**Reviewer Confidence:**

4: Quite sure. I tried to check the important points carefully. It's unlikely, though conceivable, that I missed something that should affect my ratings.

**Typos Grammar Style And Presentation Improvements:**

-- It would be helpful to include a table in the appendix with all the nitty-gritty details of implementation/optimization (e.g., hyperparameters, packages).

---

> ### Author Rebuttal · Authors · 2023-08-28
>
> Thank you very much for your encouraging feedback! Regarding your questions:
>
> > Did you spot any interesting cross-linguistic trends in your results? e.g., by language families?
>
> We did not thoroughly check cross-linguistic trends or cross-lingual transfer, this would be an interesting area for follow-up work.
>
> > Are the developed algorithmic methods relevant for other morphological tasks or other segmentation tasks?
>
> The normalisation-to-segmentation algorithms are relevant to any problem of finding subsequences with minimal total edit distance to a second set of sequences of equal length. This could for example also be useful to compute a surface-level segmentation from a canonical segmentation [1]. We will clarify this in the paper.
>
> > It would be helpful to include a table in the appendix with all the nitty-gritty details of implementation/optimization (e.g., hyperparameters, packages).
>
> We will make sure to include such a table in the final version.
>
> [1]: https://aclanthology.org/N16-1080/

---

### Official Review · Reviewer_X8iv · 2023-08-05

**Soundness:** 3

**Excitement:**

3: Ambivalent: It has merits (e.g., it reports state-of-the-art results, the idea is nice), but there are key weaknesses (e.g., it describes incremental work), and it can significantly benefit from another round of revision. However, I won't object to accepting it if my co-reviewers champion it.

**Paper Topic And Main Contributions:**

This paper proposes a two-stage approach for splitting compounds into constituent parts. The first stage consists of training a character/byte based model (ByT5 is used in the paper) in a self-supervised way over words written with hyphens as positive examples and words without hyphens as negative (non-compound) examples. Hyphens without a following new-line symbol are used as a "high precision, low recall" marker of compound-constituent boundary. The fact that many compounds are written both with and without hyphens (cf. fine-tuning vs. finetuning) helps the model to generalise. The second stage consists of further training this model in a supervised way on the new dataset based on Wiktionary.

In order to compare character-based and subword-based models, the authors use their model to make an additional pass over the pre-training dataset for subword-based langauges models, such as mT5, at the tokenisation stage. The standard SentencePiece tokeniser is applied to texts where compounds have been separated and as a result subword tokens, ideally, do not span the compound-constituent boundary. The resulting tokeniser is called CompoundPiece.

The authors show that their ByT5-based model achieves better performance on the task of compound segmentation and compound normalisation (segmentation + lemmatisation) than language-specific models from the Sigmorphon Shared Task 2022. From among the models trained by the authors, ByT5 has the best results on average, outperforming subword-based models.

A particular problem addressed by the authors is the issue of hard compounds: the compounds where the constituent boundaries do not coincide with the boundaries of subwords used by the respective tokeniser (e.g., when the word "windsurfing" is tokenised as "_winds_-ur-f-ing"). One of the aims of CompoundPiece is to reduce the number of hard compounds in training datasets. The results on this task are decent in the monolingual setting (the percentage of hard compound falls from 27.1% to 9.7%) but not great in the multilingual setting (23.2% vs. 16.5%).

**Questions For The Authors:**

In line 326, you mentioned "computed ground-truth segmentation". This looks contradictory: a computed piece of data cannot be "ground-truth" unless it is derived deterministically from some other source of ground truth. Does this mean "computed segmentation that corresponds best to ground-truth normalisation from Wiktionary"?

**Reasons To Accept:**

The paper addresses a well-defined and important task, proposes a resonable set of methods for solving it, and achieves meaningful results, most noticeably beating ad hoc language-specific models. Additionally, a new dataset for training decompounding models is proposed.

**Reasons To Reject:**

The main drawback of the paper is the complete lack of extrinsic evaluation. Better performance on compound segmentation should lead to better downstream performance, and without checks of this kind it is not evident if we should bother with retraining the models using CompoundPiece (as opposed to any of the other approaches to improving multilingual tokenisation that have been proposed in recent years). The rather pedestrian performance of the CompoundPiece on reducing the number of hard compounds in the multilingual setting makes one wonder if the proposed method generalises well: in some languages and orthographies hyphens are normally not used at all, which is an evident impediment. Generally, while the basic compound-splitting model is presented thoroughly and looks like a "finished product", its usefulness is rather narrow, while the merits of CompoundPiece remain harder to ascertain.

**Reproducibility:**

3: Could reproduce the results with some difficulty. The settings of parameters are underspecified or subjectively determined; the training/evaluation data are not widely available.

**Reviewer Confidence:**

4: Quite sure. I tried to check the important points carefully. It's unlikely, though conceivable, that I missed something that should affect my ratings.

---

> ### Author Rebuttal · Authors · 2023-08-28
>
> Thank you very much for your feedback!
>
> Although we did not run extrinsic evaluation on the usual NLU tasks, we do provide extrinsic results on Compound Segmentation and Normalisation in Table 5 (which are downstream tasks used e.g. in online dictionaries).
>
> The problem with faithfully running a more thorough extrinsic evaluation is that due to our limited computational budget we have not been able to train a new T5 model from scratch which can compete with the prior T5 models (c.f. Footnote 10). We did now run additional extrinsic experiments on WikiANN NER* obtaining 17.2% F1 score with the SentencePiece model and 24.6% F1 score with our CompoundPiece model. This is promising, but it is not clear how this improvement translates to more fully trained models.
>
> We hope to follow up with work on efficiently transferring models trained with SentencePiece (or another tokenizer) to CompoundPiece. This would allow more meaningful extrinsic evaluation on a limited budget, but would have been out of scope for this work.
>
> To answer your question:
>
> > In line 326, you mentioned "computed ground-truth segmentation". This looks contradictory: a computed piece of data cannot be "ground-truth" unless it is derived deterministically from some other source of ground truth. Does this mean "computed segmentation that corresponds best to ground-truth normalisation from Wiktionary"?
>
> This is correct, we will change the wording to reflect this fact.
>
> *Training a multilingual model jointly on the train set of all languages, and macro-averaging F1 score of the test set of each language.

---

### Meta-Review · Area_Chair_dZwo · 2023-09-24

**Recommendation:** 4

**Metareview:**

Summary (adapted from Reviewer 8bjC): This paper creates a multi-lingual data collection that contains information on compounds and their boundaries. Using the data collection, a number of systems are developed that aim to automatically identify the compound boundaries. The dataset and thus the evaluation are performed on a much wider set of languages than previous work, which tended to focus on compound-heavy languages. Wiktionary and the vocabulary of a large-scale web corpus are used as data sources.

The soundness (3/3/4) and excitement (3/4/4) scores suggest the paper is overall sound and there is excitement for its content. The discussion period identified minor clarifications that the authors should make in revisions, and that there are some desirable follow-up experiments to be run that were not feasible due to computational resource limitations. Reviewer X8iv stated that there are efficient ways to run these resources and that resource limitations should not be a barrier; the authors may want to address this in a discussion or limitations section.

As reviewer X8iv points out, there is no extrinsic (i.e. a downstream, non-decompounding task) evaluated. As reviewer WAo7 points out, it’s not clear that this approach will have any impact outside of the decompounding task.

However, this paper provides both a much larger dataset than was available through previous work, and substantial experiments, including showing that their approach even outperforms hand-tuned per-language models (Table 2). The reviews suggest that this is a substantial contribution, even if perhaps in a narrow area that does not yet have demonstrated downstream impact.

A historical note for the authors: you have correctly identified work such as Koehn and Knight (2003) in the early 2000s as the first widely known statistical unsupervised decompounding approaches. What is less widely known is that there was a series of unsupervised (and later adding semi-supervised) morphological decomposition shared tasks known as Morpho Challenge, running from 2005-2010 [1]. While they did not separately evaluate decompounding, it was key to performance, and [2] explicitly integrates Koehn and Knight’s objective into a morphological analyzer lead to then state-of-the-art results in Finnish. Whether you choose to discuss or review this work is of course, entirely up to you.

[1] http://morpho.aalto.fi/events/morphochallenge/
[2] C. Lignos, Learning from Unseen Data. Proceedings of Morpho Challenge 2010. https://aaltodoc.aalto.fi/bitstream/handle/123456789/827/isbn9789526033303.pdf?sequence=1&isAllowed=y#page=37

---

### Decision · Program_Chairs · 2023-10-07

**Decision:**

Accept-Main

**Comment:**

Summary (adapted from Reviewer 8bjC): This paper creates a multi-lingual data collection that contains information on compounds and their boundaries. Using the data collection, a number of systems are developed that aim to automatically identify the compound boundaries. The dataset and thus the evaluation are performed on a much wider set of languages than previous work, which tended to focus on compound-heavy languages. Wiktionary and the vocabulary of a large-scale web corpus are used as data sources.

The soundness (3/3/4) and excitement (3/4/4) scores suggest the paper is overall sound and there is excitement for its content. The discussion period identified minor clarifications that the authors should make in revisions, and that there are some desirable follow-up experiments to be run that were not feasible due to computational resource limitations. Reviewer X8iv stated that there are efficient ways to run these resources and that resource limitations should not be a barrier; the authors may want to address this in a discussion or limitations section.

As reviewer X8iv points out, there is no extrinsic (i.e. a downstream, non-decompounding task) evaluated. As reviewer WAo7 points out, it’s not clear that this approach will have any impact outside of the decompounding task.

However, this paper provides both a much larger dataset than was available through previous work, and substantial experiments, including showing that their approach even outperforms hand-tuned per-language models (Table 2). The reviews suggest that this is a substantial contribution, even if perhaps in a narrow area that does not yet have demonstrated downstream impact.

A historical note for the authors: you have correctly identified work such as Koehn and Knight (2003) in the early 2000s as the first widely known statistical unsupervised decompounding approaches. What is less widely known is that there was a series of unsupervised (and later adding semi-supervised) morphological decomposition shared tasks known as Morpho Challenge, running from 2005-2010 [1]. While they did not separately evaluate decompounding, it was key to performance, and [2] explicitly integrates Koehn and Knight’s objective into a morphological analyzer lead to then state-of-the-art results in Finnish. Whether you choose to discuss or review this work is of course, entirely up to you.

[1] http://morpho.aalto.fi/events/morphochallenge/
[2] C. Lignos, Learning from Unseen Data. Proceedings of Morpho Challenge 2010. https://aaltodoc.aalto.fi/bitstream/handle/123456789/827/isbn9789526033303.pdf?sequence=1&isAllowed=y#page=37